

Figure 3: Word cloud of key terms from the questions, illustrating the diversity of compositional content evaluated in the benchmark.

# A    Appendix

## A.1    Justification of the 13-Category Taxonomy

**Operationalizing compositionality.** In MMComposition, a sample is *compositional* if correctly answering it requires jointly processing at least two distinct visual-semantic primitives from the set {object identity, object attributes, spatial/semantic relations, quantity, cross-image correspondence}, where no single primitive alone suffices for the answer. Table 8 maps each of our 13 categories to the specific primitives it requires.

Table 8: Required visual-semantic primitives per category.

| Category | Required Primitives |
|---|---|
| Attr-P / Attr-R | object identity + attributes |
| Obj-P / Obj-R | object identity + attributes + relations |
| Count-P / Count-R | object identity + quantity |
| Rel-P / Rel-R | object identity + spatial/semantic relations |
| Obj-Interact | object identity + relations + attributes |
| Diff-S | cross-image corresp. + attributes/relations |
| Visual-Sim | cross-image corresp. + object identity |
| TR | object identity + textual attributes |
| Probing | all primitives (multi-correct) |

**Design rationale.** Our 13 evaluation categories were developed through systematic analysis of the compositional challenges present in our seed datasets, guided by but extending beyond the coarse-grained dimensions (object, attribute, relation) adopted by prior compositionality benchmarks. The design was informed by two key observations from existing work:

First, SugarCrepe (Hsieh et al., 2024) organizes compositionality evaluation by perturbation type (Replace, Swap, Add) applied to three atomic concepts (object, attribute, relation), yielding 7 fine-grained hard negative types—all evaluated via binary image-to-text retrieval. Second, CREPE (Ma et al., 2023) evaluates compositionality along two cognitive dimensions—systematicity and productivity—with complexity-graded hard negatives (atomic, swap, negate), again restricted to retrieval tasks. While both benchmarks provide

valuable insights into basic compositional discrimination, they share three fundamental limitations: (1) they only test whether models can distinguish correct from incorrect captions for a single image, without requiring multi-step reasoning; (2) they do not evaluate counting, cross-image comparison, or object interaction; and (3) they use binary selection rather than multi-correct formats that probe the robustness of compositional judgment.

To address these gaps, we designed MMComposition's taxonomy around three principles:

**Perception–Reasoning separation.** For each core semantic dimension (object, attribute, relation, counting), we distinguish perception tasks (recognizable "within a blink") from reasoning tasks (requiring multi-step inference). This separation reveals that a model may perceive individual compositional facts correctly yet fail when sequential reasoning is needed—a distinction invisible to existing benchmarks.

**Counting as a distinct compositional skill.** Counting requires aggregating and individuating objects across a scene, often in the presence of occlusion, similar distractors, and irregular arrangements. No prior compositionality benchmark isolates this skill, yet it represents a major failure mode for current VLMs (see Table 2: GPT-5 achieves only 75.86% on Count-P vs. human 93.06%).

**Five novel categories.** We introduce Object Interaction (functional multi-object relations), Difference Spotting (cross-image compositional comparison), Visual Similarity (multi-image similarity judgment), Text Rendering (reading text within visual contexts), and Compositional Probing (multi-correct selection). These categories test compositional capabilities that are qualitatively different from the basic discrimination tasks in existing benchmarks.    **Evaluation coverage comparison.** Table 9 compares MMComposition's

Table 9: Evaluation coverage comparison across VL compositionality benchmarks. ✓ = directly evaluated as a dedicated category; △ = partially overlapping (the dataset touches this dimension but not as a targeted evaluation); ✗ = not covered.

| Dataset | Task Format | Perception | | | | | | | Reasoning | | | | | Probing |
|---|---|---|---|---|---|---|---|---|---|---|---|---|---|---|
| | | Obj-P | Attr-P | Count-P | Rel-P | Diff-S | TR | Vis-Sim | Attr-R | Obj-R | Count-R | Rel-R | Obj-Int | Prob |
| ARO | Binary I2T Retr. | △ | △ | ✗ | △ | ✗ | ✗ | ✗ | ✗ | ✗ | ✗ | ✗ | ✗ | ✗ |
| SugarCrepe | Binary I2T Retr. | △ | △ | ✗ | △ | ✗ | ✗ | ✗ | ✗ | ✗ | ✗ | ✗ | ✗ | ✗ |
| CREPE | I2T Retrieval | △ | △ | ✗ | △ | ✗ | ✗ | ✗ | ✗ | ✗ | ✗ | ✗ | ✗ | ✗ |
| VL-Checklist | Binary I2T Retr. | △ | △ | ✗ | △ | ✗ | ✗ | ✗ | ✗ | ✗ | ✗ | ✗ | ✗ | ✗ |
| Cola | T2I Retrieval | △ | △ | ✗ | ✗ | ✗ | ✗ | ✗ | ✗ | ✗ | ✗ | ✗ | ✗ | ✗ |
| Winoground | I2T + T2I Match | △ | △ | ✗ | △ | ✗ | ✗ | ✗ | ✗ | △ | ✗ | △ | ✗ | ✗ |
| FineMatch | Mismatch Detec. | △ | △ | ✗ | △ | ✗ | ✗ | ✗ | ✗ | ✗ | ✗ | ✗ | ✗ | ✗ |
| GQA | Compositional QA | △ | △ | △ | △ | ✗ | ✗ | ✗ | △ | △ | △ | △ | ✗ | ✗ |
| **MMComposition** | **Comp. QA** | ✓ | ✓ | ✓ | ✓ | ✓ | ✓ | ✓ | ✓ | ✓ | ✓ | ✓ | ✓ | ✓ |

evaluation coverage with existing VL compositionality benchmarks. We use three levels of coverage: ✓ (directly and comprehensively evaluated), △ (partially overlapping—the dataset touches this dimension but not as a targeted evaluation), and ✗ (not covered). The comparison reveals that prior benchmarks predominantly evaluate along the object/attribute/relation perception axis, while MMComposition is the first compositionality benchmark to systematically cover counting, cross-image comparison, object interaction, and multi-correct probing.

## A.2 Definition of 13 distinct categories in MMComposition

- **Attribute Perception**: The specific attributes or properties of the object perception task that can be solved by humans "within a blink".

- **Object Perception**: Identification or recognition of objects in the image.

- **Counting Perception**: Counting the number of objects or elements in the image.

- **Relation Perception**: Understanding the relationships between objects in the image.

- **Difference Spotting**: Identifying differences or changes between objects or scenes in two similar images.

- **Text Rendering**: Reading or interpreting text present in the image.

- **Visual Similarity**: Comparing similarities between objects or elements across multiple images.

- **Attribute Reasoning**: Identifying and reasoning about specific attributes or properties of objects in the image.

- **Object Reasoning**: Identifying and reasoning about objects in the image.

- **Counting Reasoning**: Identifying and reasoning about the number of objects or elements in the image.

- **Relation Reasoning**: Identifying and reasoning about the spatial arrangement or positioning of objects in the image.

- **Object Interaction**: Understanding interactions among multiple objects in the image.

- **VL Composition Probing**: Examining the composition or combination of visual and textual elements in images, where models are required to accurately find all the complex compositional descriptions about the image.

### A.3 Quantitive Results of MMComposition

In this section, we show statistical results for MMCOMPOSITION in Figure 5 through Figure 7.

### A.4 Comparison Analysis of Image Encoding in GPT-4o, Qwen2-VL, and InternVL-2

In GPT-4o, when the image detail parameters are set to "high", images are first scaled to fit within a $2048 \times 2048$ square while maintaining their aspect ratio. Then, the images are further scaled so that the shortest side is 768px long. Finally, GPT-4o calculates how many 512px squares the image contains, with each square costing 170 tokens. An additional 85 tokens for low resolution are always added to the final total. As a result, GPT-4o does not achieve true "any resolution" image processing.

In Qwen2-VL and InternVL-2, the image encoders adopt a dynamic "any resolution" encoding strategy. The images are first mapped to an optimal aspect ratio from predefined ratios, then divided into $448 \times 448$ or $28 \times 28$ pixel tiles, with each tile converted into 256 or 1 image tokens. A thumbnail is then generated to capture the global context. This allows the encoders to handle images of any resolution and aspect ratio. Furthermore, the image encoder in Qwen2-VL is a 675M ViT with a two-dimensional positional encoding mechanism, while InternVL-2 utilizes the more powerful InternViT with 6B parameters. This distinction contributes to the superior performance of the compositionality of Qwen2-VL and InternVL-2 in our benchmark. In Table 10, we provide a comparison of the properties of visual encoders for the aforementioned models.

Table 10: Visual encoder comparison of GPT-4o, InternVL2 and Qwen2-VL.

| Method | Visual Encoder | Image Tile Size | Maximum Number of Tiles | Maximum Aspect Ratio | # of Tokens for One Tile |
|---|---|---|---|---|---|
| GPT-4o | - | 512 x 512 | 8 | any | 170 |
| InternVL2 | InternViT-6B | 448 x 448 | 12 | 1:6 | 256 |
| Qwen2-VL | ViT-675M | 28 x 28 | dynamic | any | 1 |

### A.5 Characteristics of Questions Where Models Underperform

We define the comprehensive performance value (CPV) for each question as the average score across 54 representative VLMs. By comparing each question's CPV with the score of a random choice within its class, we find that 1,159 questions have a CPV lower than that of random chance. We show statistical results questions with low CPV in Figure 9 through Figure 12.

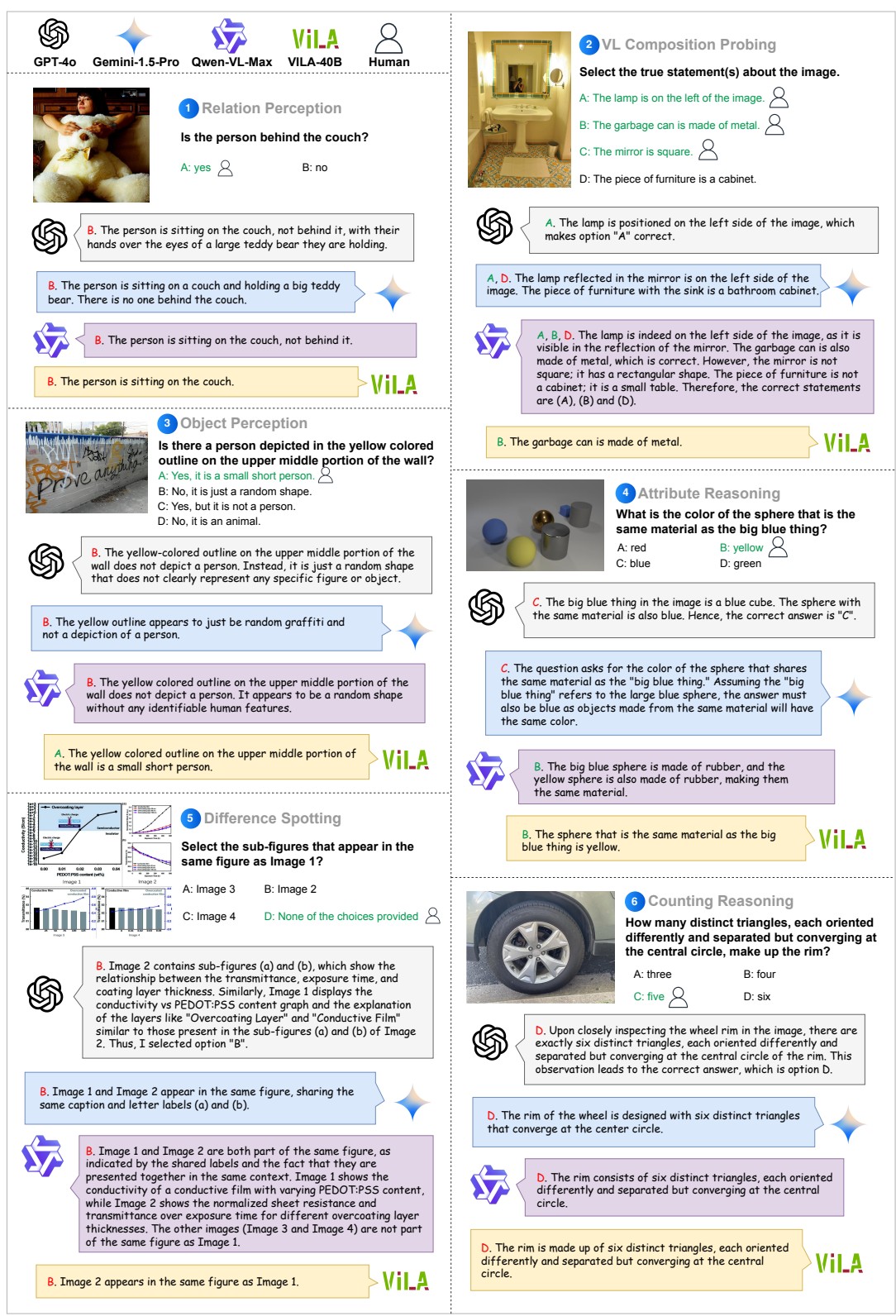

Figure 4: Interpretable analysis of different VLMs. Green letters indicate correct answers, while red letters represent wrong (predicted) answers.

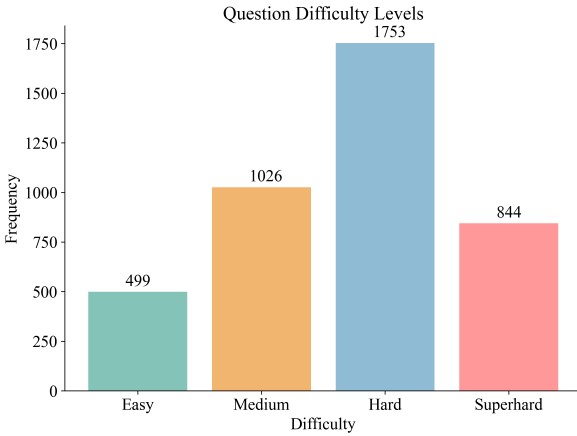

Figure 5: Distribution of difficulty levels across the question set, illustrating the challenging nature of tasks.

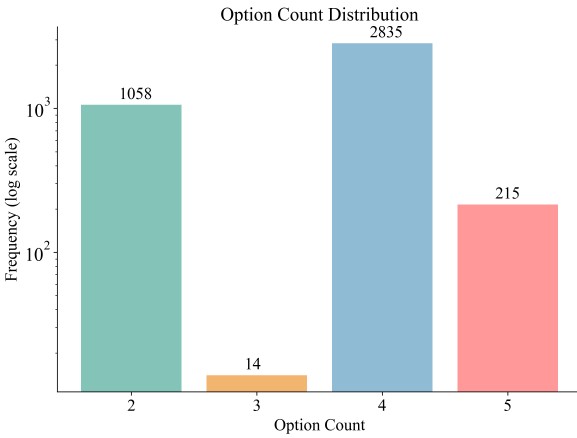

Figure 6: Distribution of option counts per question, showing the variety in answer choices provided to evaluate VLMs.

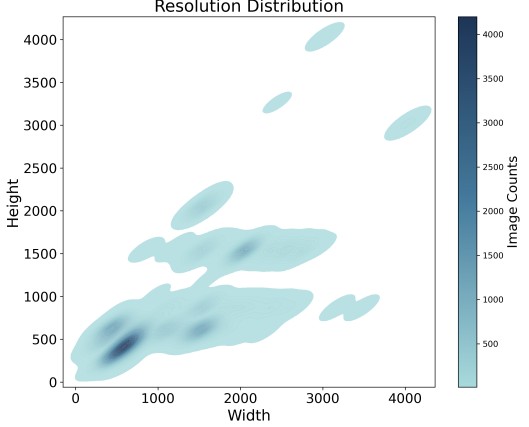

Figure 7: Resolution distribution of images in our benchmark, reflecting the portion of high-quality images in MMCOMPOSITION.

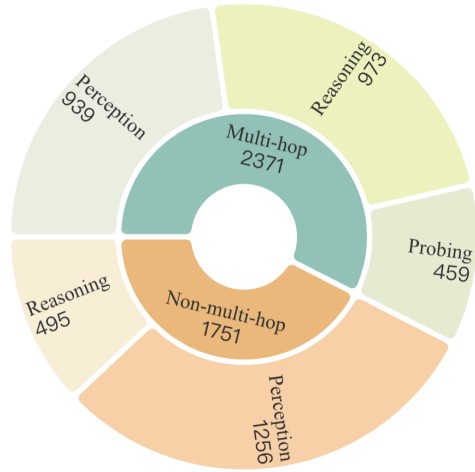

Figure 8: Distribution of multi-hop and non-multi-hop questions across categories.

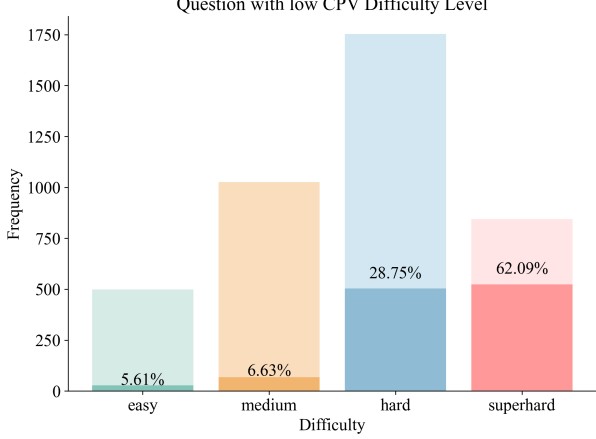

Figure 9: Distribution of difficulty levels of questions with low CPV, illustrating the authenticity of the difficulty distribution in MMCOMPOSITION.

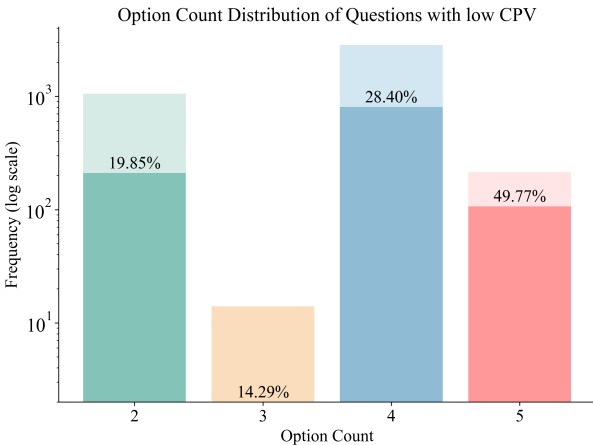

Figure 10: Distribution of option counts for questions with low CPV.

Table 11: Comparison with related VL benchmarks: "Multi-Hop" refers to whether the dataset contains questions that need multi-hop reasoning, "Comprehensive" in the Capabilities column indicates the benchmark evaluates multiple capabilities for VLMs (e.g., recognition, OCR, knowledge, math, and spatial reasoning).

| Dataset | Size | Human Annotation | Multi-Hop | Capabilities | Best Performance (Model/Human) |
|---|---|---|---|---|---|
| MMBench Liu et al. (2023b) | 3,217 | ✗ | ✗ | Comprehensive | 86.1 / - |
| MME Fu et al. (2023) | 2,800 | ✓ | ✗ | Comprehensive | 1790.04/- |
| MMStar Chen et al. (2024a) | 1,500 | ✓ | ✗ | Comprehensive | 66.0/- |
| SeedBench Liu et al. (2023b) | 19k | ✓ | ✗ | Comprehensive | 72.4 / - |
| MMMU Yue et al. (2023) | 11.5k | ✓ | ✗ | College-Level Subject Knowledge | 69.1 / 88.6 |
| HalBench Guan et al. (2024) | 1,129 | ✓ | ✗ | Hallucination | 67.58 / - |
| **MMComposition (ours)** | 4,122 | ✓ | ✓ | Compositionality | 68.16 / 90.31 |

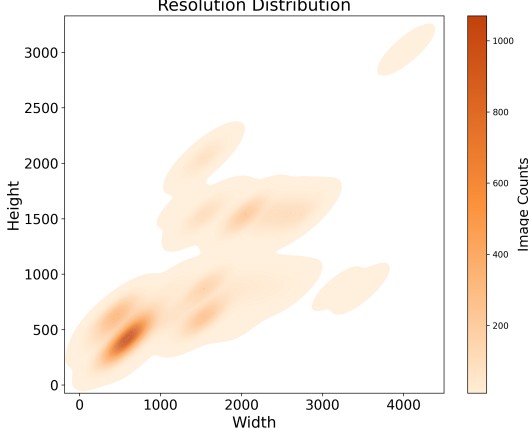

Figure 11: Resolution distribution of images with low CPV.

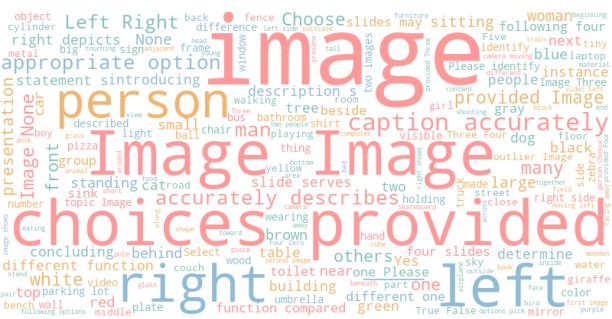

Figure 12: Word cloud of key terms from the questions with low CPV, illustrating the keywords appearing in the questions that VLMs are hard to answer currently.

### A.6 Analysis of GPT-4o's Underperformance in Specific Tasks

Since GPT-4o performs relatively weaker on Obj-P, Count-P, Attr-R, Count-R, and Rel-R tasks compared to smaller models that outperform it, we aim to provide an intuitive analysis of the reasons behind its poor performance on these tasks. Figure 13 presents interpretable examples for the aforementioned categories.

### A.7 More Interpretable Examples

To provide a clearer and more comprehensive interpretation of the models' capabilities, we present additional interpretable examples in Figure 15 and Figure 16.

### A.8 Additional Experiments

To further verify the challenging nature of MMCOMPOSITION and demonstrate the indispensable role of images, we conducted additional experiments under image-blind settings. The results are presented in Table 12. We also conducted experiments to compare different visual-to-language (V2L) adapters and their impact on model performance, as summarized in Table 13. Additionally, we examined the models' abilities to handle multi-hop reasoning questions and the performance when multiple images are used. These experiments aim to provide a comprehensive understanding of the factors influencing model performance on MMCOMPOSITION.

**Image-blind Setting.** As shown in Table 12, all models experienced significant performance drops across all evaluation dimensions when visual inputs were removed. For instance, the overall score of Qwen2-VL-72B decreased by **22.74**, underscoring the indispensable role of images in these tasks. This substantial decline confirms that MMCOMPOSITION effectively evaluates the integration of visual and linguistic understanding, as models struggle without visual context. The overall scores of GPT-3.5 and LLaMA-3.1-70B are only

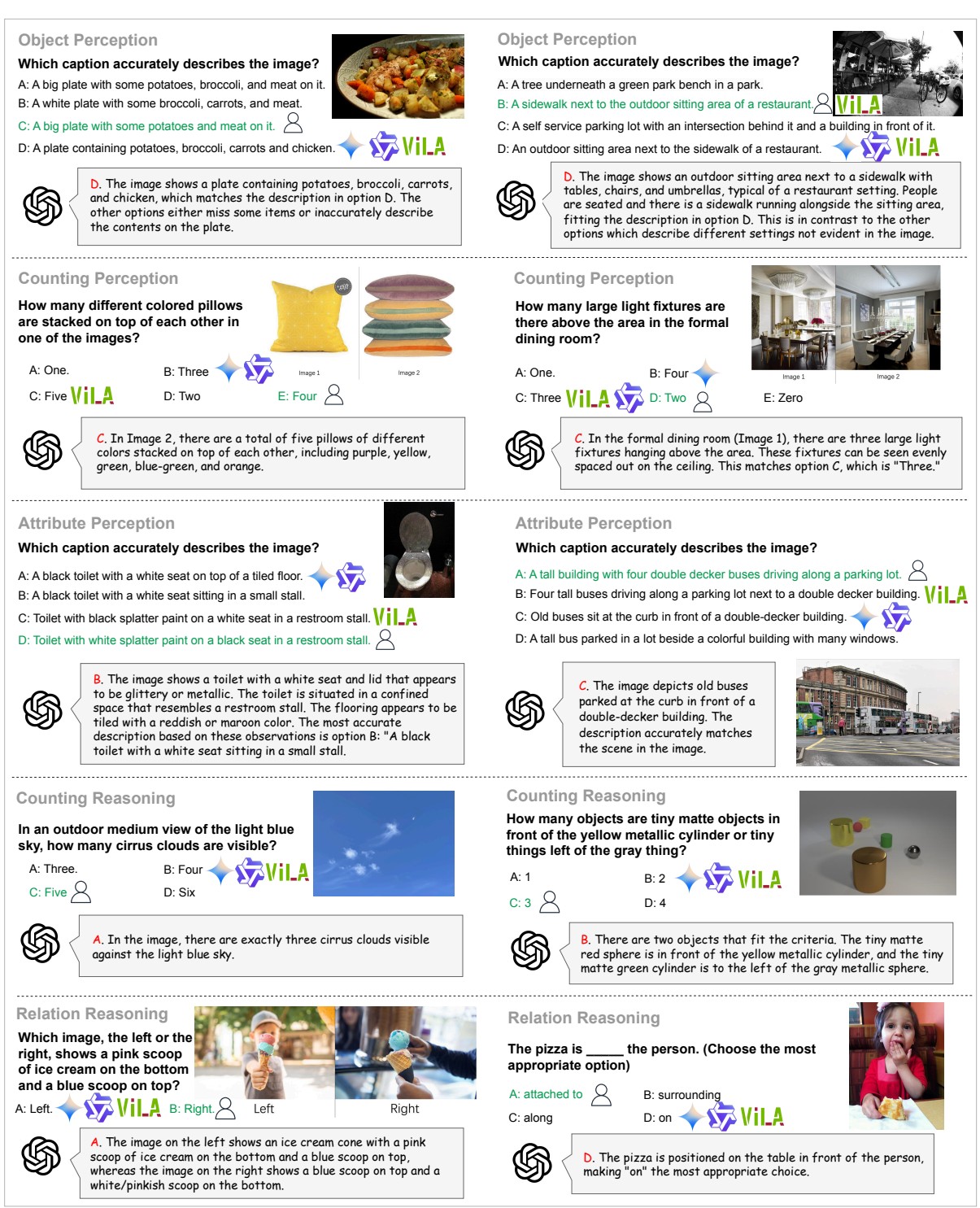

Figure 13: GPT-4o Weak Category Analysis. The logos of the models or human displayed to the right of the option(s) indicate that the model or human has selected the option(s) as the correct answer(s).

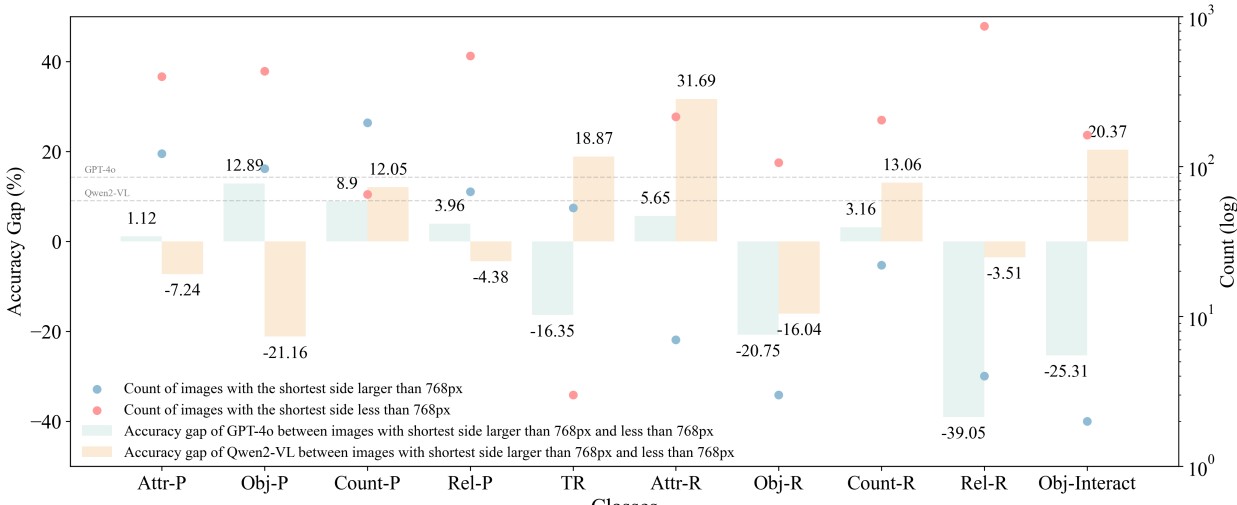

Figure 14: Performance gap between images whose shortest side > 768px and those ≤ 768px, defined as gap = $Acc_{>768px} - Acc_{\leq 768px}$. The histogram shows the distribution of performance gaps across 13 tasks. The average performance gap for GPT-4o is 14.26, while for Qwen2-VL, it is 9.05. The smaller gap for Qwen2-VL indicates its greater effectiveness in processing high-resolution images. Additionally, Qwen2-VL's performance gaps are more consistently positive across different tasks, further highlighting its robustness in handling high-resolution images.

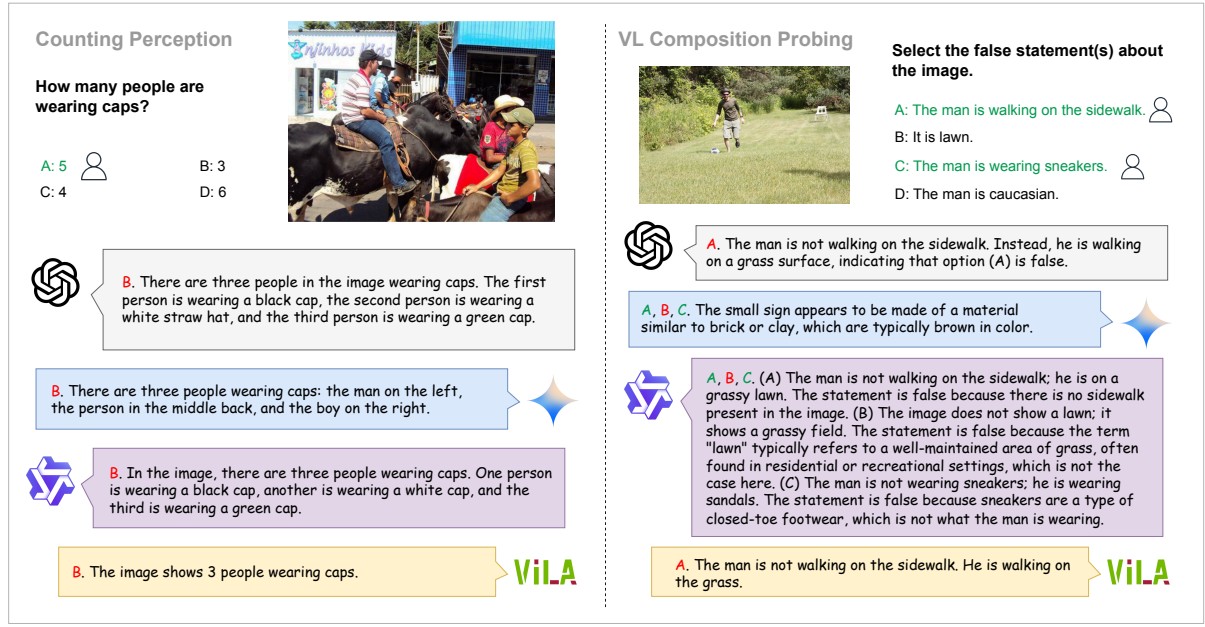

Figure 15: More interpretable analysis of different VLMs. Green indicates correct answers, while red represents the predicted wrong answers.

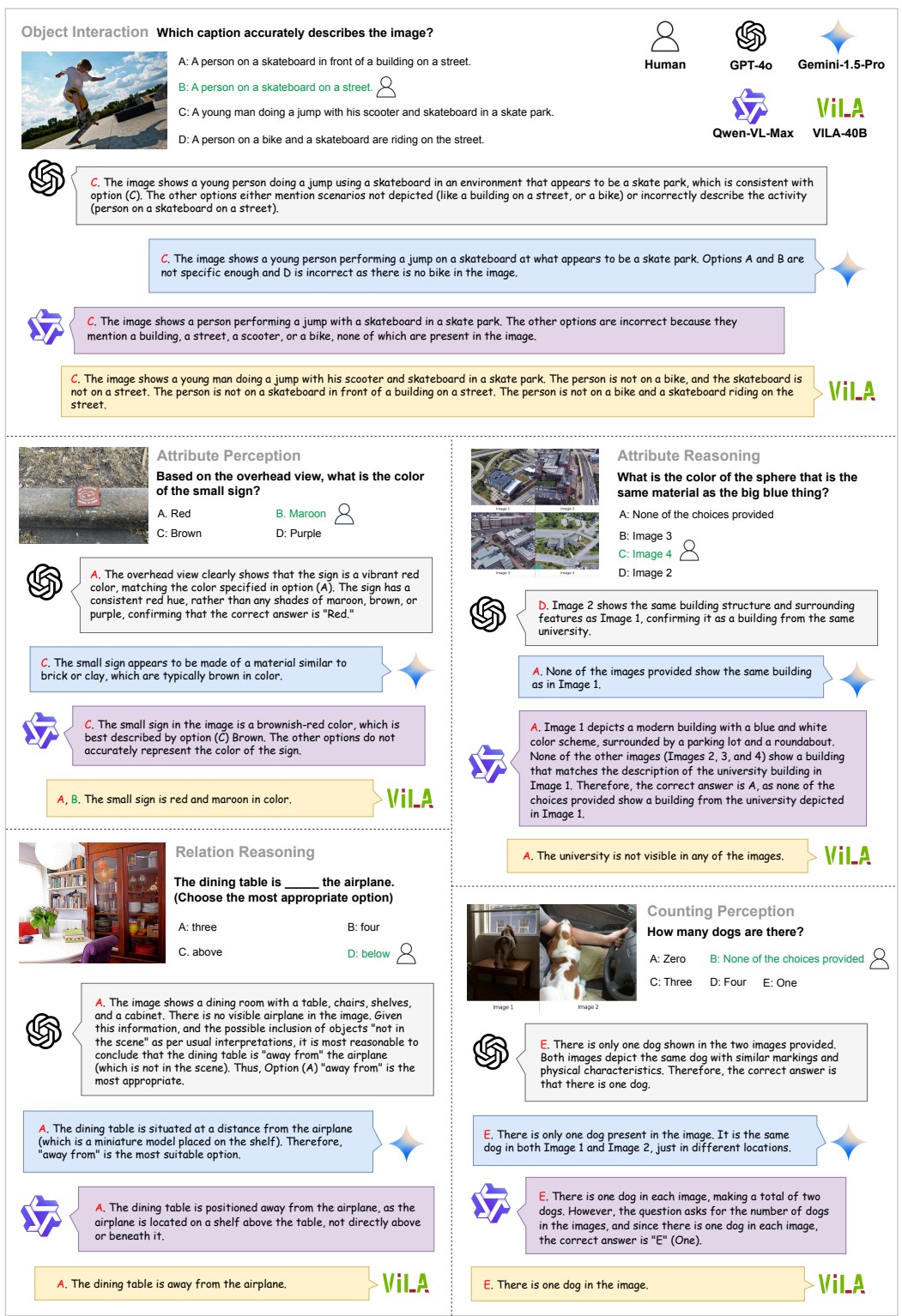

Figure 16: More interpretable analysis of different VLMs. Green indicates correct answers, while red represents the predicted wrong answers.

slightly higher than that of a random choice, indicating that our questions cannot be answered using common sense.

**V2L Adapters Comparison.** We also compare different V2L adapters. As shown in Table 13, models that utilize an `MLP` adapter (e.g., LLaVA1.5-13B) generally outperform those with a `Q-Former` adapter in overall performance. Specifically, LLaVA1.5-13B achieves an overall score of **41.02**, surpassing InstructBLIP-13B's score of **36.76**. This suggests that the choice of adapter architecture significantly influences a model's ability to effectively integrate visual features.

**Multi-hop/Non-multi-hop Question Setting.** We analyze the performance on multi-hop versus non-multi-hop question settings in Table 14 and observe that some models perform better on multi-hop questions, indicating strength in complex reasoning tasks. For example, InternVL2-40B achieved an overall score of **62.95** on multi-hop questions, compared to **73.35** on non multi-hop ones. This demonstrates the model's enhanced capability to handle questions requiring multiple reasoning steps.

**Multi-image Setting.** As shown in Table 15, models respond differently to multiple images. Qwen2-VL-72B improves its overall score by **4.57**, indicating effective use of additional visual evidence, whereas InternVL2-40B drops by **1.63**, suggesting challenges in integrating information across images. These additional experiments underscore the difficulty of MMCOMPOSITION and highlight the role of visual evidence, adapter design, and the challenges of multi-hop reasoning in multimodal models. They also offer guidance for future work on improving vision–language model capabilities.

Table 12: Results for Image-Blind Setting using VLMs and LLMs.

| Model | Perception | Reasoning | Probing | Overall |
|---|---|---|---|---|
| Qwen2-VL-72B | 59.67 | 76.77 | 69.57 | 66.86 |
| Qwen2-VL-72B-blind | $44.72_{-14.95}$ | $47.41_{-29.36}$ | $30.76_{-38.81}$ | $44.12_{-22.74}$ |
| InternVL2-26B | 59.51 | 69.48 | 52.43 | 62.27 |
| InternVL2-26B-blind | $33.97_{-25.54}$ | $42.30_{-27.18}$ | $32.17_{-20.26}$ | $36.74_{-25.53}$ |
| InternVL2-40B | 64.55 | 74.66 | 59.59 | 67.60 |
| InternVL2-40B-blind | $37.46_{-27.09}$ | $43.66_{-31.00}$ | $34.28_{-25.31}$ | $39.32_{-28.28}$ |
| InternVL2-76B | 62.56 | 75.34 | 58.46 | 66.65 |
| InternVL2-76B-blind | $33.24_{-29.32}$ | $43.26_{-32.08}$ | $32.68_{-25.78}$ | $36.74_{-29.91}$ |
| GPT-3.5 | 26.87 | 41.08 | 32.93 | 32.61 |
| LLaMA-3.1-70B | 36.11 | 35.35 | 26.58 | 34.78 |

Table 13: Comparison of Different Adapters for Model's Performance.

| Model | Visual Encoder | LLM | V2L Adapter | Perception | Reasoning | Probing | Overall |
|---|---|---|---|---|---|---|---|
| mPLUG-Owl2 Ye et al. (2024) | ViT-L/14 | LLaMA2-7B | Q-Former | 36.26 | 45.16 | 30.36 | 38.77 |
| InstructBLIP-7B Dai et al. (2023) | ViT-G/14 | Vicuna-7B | Q-Former | 33.23 | 43.05 | 31.41 | 36.53 |
| LLaVA1.5-7B Liu et al. (2024b) | ViT-L/14 | Vicuna-7B | MLP | 35.76 | 44.28 | 30.32 | 38.19 |
| InstructBLIP-13B Dai et al. (2023) | ViT-G/14 | Vicuna-13B | Q-Former | 35.51 | 42.23 | 25.24 | 36.76 |
| LLaVA1.5-13B Liu et al. (2024b) | ViT-L/14 | Vicuna-13B | MLP | 36.79 | 47.89 | 39.32 | 41.02 |

Table 14: Comparison of models' performance on multi-hop and non multi-hop questions.

| Model | Perception | Reasoning | Probing | Overall |
|---|---|---|---|---|
| InternVL2-40B-non-multi-hop | 72.56 | 75.35 | - | 73.35 |
| InternVL2-40B-multi-hop | 52.35 | 74.31 | 59.59 | 62.95 |
| Qwen2-VL-72B-non-multi-hop | 59.49 | 76.97 | - | 64.43 |
| Qwen2-VL-72B-multi-hop | 58.81 | 76.46 | 69.57 | 68.31 |
| VILA-40B-non-multi-hop | 65.54 | 69.29 | - | 66.60 |
| VILA-40B-multi-hop | 44.40 | 68.14 | 62.16 | 57.82 |
| GPT-4o-non-multi-hop | 62.30 | 63.03 | - | 62.51 |
| GPT-4o-multi-hop | 48.95 | 63.00 | 54.65 | 55.94 |
| LLaVA-1.6-34B-non-multi-hop | 65.27 | 68.69 | - | 66.24 |
| LLaVA-1.6-34B-multi-hop | 44.06 | 51.90 | 58.17 | 50.12 |
| Gemini-1.5-Pro-non-multi-hop | 54.71 | 53.54 | - | 54.38 |
| Gemini-1.5-Pro-multi-hop | 42.48 | 58.99 | 49.60 | 50.78 |

Table 15: Results for Multi-image Setting.

| Model | Perception | Reasoning | Probing | Overall |
|---|---|---|---|---|
| Qwen2-VL-72B | 55.36 | 74.68 | 89.86 | 70.09 |
| Qwen2-VL-72B-multi | $63.01_{+7.65}$ | $78.55_{+3.87}$ | $89.19_{-0.67}$ | $74.66_{+4.57}$ |
| InternVL2-40B | 42.35 | 71.45 | 88.51 | 63.79 |
| InternVL2-40B-multi | $39.29_{-3.06}$ | $70.81_{-0.64}$ | $86.49_{-2.02}$ | $62.16_{-1.63}$ |

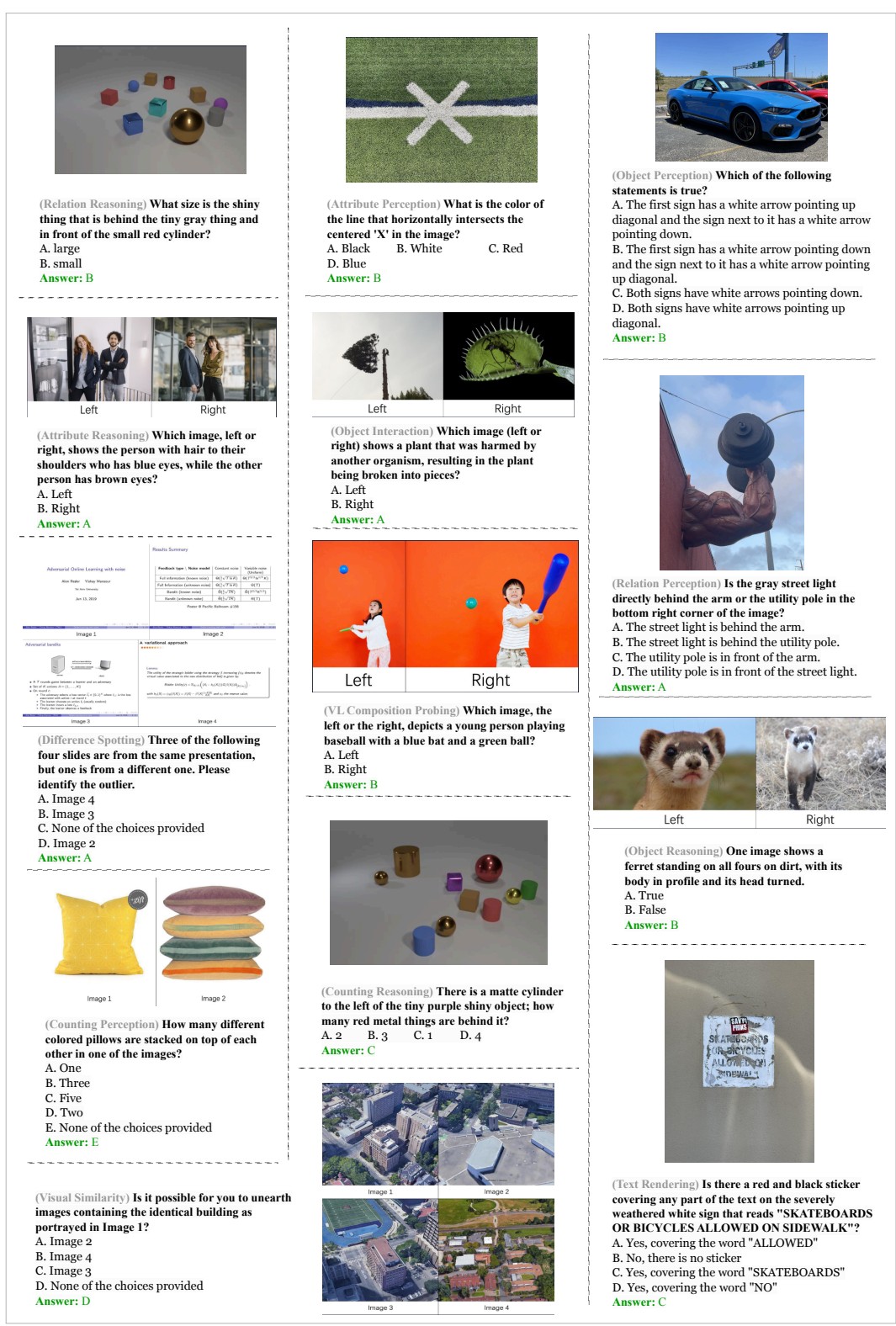

Figure 17: Examples of multi-hop questions: The ratio of multi-hop to non-multi-hop questions in our dataset is 2,371 to 1,751.

Table 16: The comprehensive performance of 93 VLMs on Acc, including open source models and API-based models . The **best** and second best results are in bold and underlined, respectively.

| Method | Perception↑ | | | | | | | Reasoning↑ | | | | | Probing↑ | Overall ↑ |
|---|---|---|---|---|---|---|---|---|---|---|---|---|---|---|
| | Attr-P | Obj-P | Count-P | Rel-P | Diff-S | TR | Visual-Sim | Attr-R | Obj-R | Count-R | Rel-R | Obj-Interact | Prob | |
| Human | 97.94 | 98.04 | 93.06 | 92.00 | 79.02 | 85.71 | 86.54 | 91.20 | 78.83 | 100.00 | 77.35 | 88.00 | 91.84 | 90.31 |
| Qwen2.5-VL-72B-Instruct (team, 2024) | **74.05** | **77.39** | **55.56** | 68.01 | 39.00 | 50.00 | 63.51 | **89.19** | **86.24** | **87.61** | **73.90** | 74.39 | 42.92 | **68.16** |
| InternVL2-40B (Chen et al., 2024b) | 72.22 | 75.99 | 45.21 | **72.53** | 31.12 | 73.21 | 48.65 | 83.78 | 82.57 | 84.51 | 69.75 | 65.85 | 59.59 | 67.60 |
| Qwen2-VL-72B-Instruct (team, 2024) | 59.57 | 63.87 | 52.49 | 62.52 | **45.23** | **82.14** | 67.57 | 87.84 | 84.40 | 84.51 | 71.49 | 70.12 | **69.57** | 66.86 |
| InternVL2-76B (Chen et al., 2024b) | 70.65 | 75.52 | 48.28 | 70.00 | 19.09 | 78.57 | 48.65 | 85.14 | 83.49 | 85.40 | 70.01 | 67.07 | 58.46 | 66.65 |
| InternVL2.5-78B-MPO (Chen et al., 2024b) | 73.28 | 73.43 | 53.64 | 67.25 | 34.85 | 50.00 | **74.32** | 87.39 | 84.40 | 85.40 | 69.61 | 70.12 | 37.98 | 65.61 |
| InternVL2.5-38B-MPO (Chen et al., 2024b) | 71.55 | 72.03 | 51.34 | 66.99 | 39.83 | 44.64 | **74.32** | 84.68 | 85.32 | 86.73 | 68.41 | 70.12 | 41.00 | 65.34 |
| InternVL-Chat-V1.2-Plus (Chen et al., 2024b) | 69.81 | 65.73 | 43.68 | 69.02 | 31.12 | 78.57 | 28.38 | 78.83 | 77.98 | 80.53 | 66.27 | 60.98 | 65.80 | 64.58 |
| InternVL2.5-26B-MPO (Chen et al., 2024b) | 69.33 | 71.10 | 47.13 | 64.15 | 27.39 | 50.00 | 51.35 | 85.14 | 81.65 | 84.96 | 70.95 | 64.63 | 37.58 | 62.90 |
| InternVL2.5-78B (Chen et al., 2024b) | 70.07 | 66.90 | 47.13 | 64.23 | 32.37 | 48.21 | 60.81 | 85.59 | 82.57 | 80.09 | 68.27 | 73.17 | 37.58 | 62.64 |
| InternVL2-26B (Chen et al., 2024b) | 68.46 | 67.13 | 40.23 | 66.96 | 22.82 | 80.36 | 62.16 | 79.28 | 79.82 | 81.86 | 62.65 | 63.41 | 52.43 | 62.27 |
| Qwen2-VL-7B-Instruct (team, 2024) | 68.30 | 71.79 | 41.38 | 64.63 | 32.37 | 39.29 | 52.70 | 81.08 | 76.15 | 80.53 | 67.34 | 69.51 | 41.43 | 62.09 |
| VILA-40B (Lin et al., 2024) | 65.70 | 64.10 | 45.21 | 63.65 | 23.65 | 75.00 | 44.59 | 70.72 | 77.06 | 67.26 | 69.08 | 59.15 | 62.16 | 61.83 |
| InternVL2.5-38B (Chen et al., 2024b) | 66.51 | 67.60 | 46.74 | 60.28 | 30.29 | 53.57 | 59.46 | 84.23 | 83.49 | 80.97 | 65.19 | 71.95 | 41.68 | 61.43 |
| InternVL2.5-26B (Chen et al., 2024b) | 67.44 | 68.53 | 43.68 | 61.99 | 21.58 | 41.07 | 51.35 | 84.68 | 81.65 | 82.30 | 66.13 | 61.59 | 39.58 | 60.45 |
| Ovis1.6-Gemma-27B (Lu et al., 2024) | 66.25 | 61.07 | 49.04 | 60.13 | 28.22 | 42.86 | 54.05 | 81.53 | 80.73 | 80.53 | 68.81 | 57.93 | 41.14 | 60.27 |
| Qwen2.5-VL-7B-Instruct (team, 2024) | 69.91 | 66.90 | 43.30 | 59.74 | 22.41 | 41.07 | 48.65 | 82.88 | 81.65 | 80.09 | 64.39 | 68.29 | 40.41 | 60.06 |
| Molmo-72B (Deitke et al., 2024) | 76.08 | 68.53 | 48.28 | 65.20 | 25.31 | 51.79 | 43.24 | 65.77 | 75.23 | 62.39 | 58.63 | 73.78 | 41.47 | 59.59 |
| InternVL-Chat-V1.2 (Chen et al., 2024b) | 64.58 | 62.00 | 41.38 | 62.98 | 25.73 | 76.79 | 29.73 | 63.06 | 71.56 | 61.06 | 63.19 | 65.24 | 60.71 | 59.13 |
| GPT-4o (Achiam et al., 2023) | 63.97 | 57.58 | 37.93 | 66.76 | 32.37 | **82.14** | 60.81 | 62.61 | 79.82 | 61.95 | 58.37 | **75.00** | 54.65 | 59.03 |
| POINTS1.5-7B-Chat (Liu et al., 2024c) | 70.13 | 61.54 | 39.46 | 60.39 | 24.90 | 46.43 | 44.59 | 76.13 | 77.06 | 76.11 | 60.24 | 69.51 | 45.21 | 58.66 |
| InternVL-Chat-V1.5 (Chen et al., 2024b) | 59.44 | 58.97 | 38.31 | 60.47 | 21.58 | 76.79 | 51.35 | 77.93 | 83.49 | 78.32 | 59.57 | 63.41 | 57.01 | 58.64 |
| InternVL2.5-8B-MPO (Chen et al., 2024b) | 65.64 | 66.20 | 45.21 | 58.32 | 21.99 | 41.07 | 56.76 | 78.83 | 80.73 | 76.55 | 65.19 | 59.76 | 37.44 | 58.49 |
| Qwen2.5-VL-3B-Instruct (team, 2024) | 64.80 | 65.03 | 40.61 | 58.98 | 29.46 | 46.43 | 35.14 | 76.58 | 77.06 | 78.76 | 61.45 | 60.37 | 39.58 | 57.71 |
| Ovis1.6-Gemma2-9B (Lu et al., 2024) | 65.16 | 61.54 | 40.61 | 55.20 | 23.65 | 41.07 | 54.05 | 80.18 | 77.06 | 76.99 | 66.13 | 67.68 | 38.63 | 57.88 |
| InternVL2-8B (Chen et al., 2024b) | 62.68 | 59.21 | 31.80 | 59.54 | 25.31 | 73.21 | 33.78 | 78.83 | 75.23 | 73.89 | 60.37 | 62.20 | 54.10 | 57.76 |
| LLaVA-V1.6-34B (Liu et al., 2024a) | 67.24 | 66.90 | 44.06 | 61.31 | 25.73 | 76.79 | 21.62 | 53.15 | 67.89 | 53.10 | 59.44 | 54.27 | 58.17 | 57.28 |
| Llama-3.2-90B-Vision-Instruct | 68.85 | 69.46 | 39.85 | 62.87 | 23.65 | 53.57 | 41.89 | 64.86 | 69.72 | 54.87 | 57.56 | 64.63 | 46.84 | 57.23 |
| InternVL2.5-8B (Chen et al., 2024b) | 64.10 | 61.07 | 39.85 | 59.23 | 19.09 | 39.29 | 36.49 | 71.62 | 73.39 | 76.11 | 63.45 | 62.20 | 40.41 | 56.54 |
| MiniCPM-V2.6 (Yao et al., 2024) | 65.19 | 58.04 | 41.00 | 61.80 | 21.99 | 73.21 | 37.84 | 63.96 | 73.39 | 68.14 | 52.07 | 60.98 | 54.43 | 56.07 |
| InternVL2.5-4B-MPO (Chen et al., 2024b) | 59.67 | 64.80 | 42.91 | 58.40 | 19.92 | 33.93 | 40.54 | 72.52 | 74.31 | 69.47 | 60.78 | 59.76 | 41.58 | 55.74 |
| InternLM-XComposer2-4KHD-7B (Dong et al., 2024b) | 62.24 | 55.24 | 39.08 | 58.36 | 23.65 | 67.86 | 27.03 | 70.72 | 74.31 | 60.18 | 55.82 | 59.15 | 60.02 | 55.35 |
| Qwen-VL-Max (Bai et al., 2023) | 53.76 | 53.15 | 36.40 | 58.67 | 22.82 | 80.36 | 41.89 | 53.60 | 65.14 | 53.98 | 60.91 | 62.80 | 63.87 | 54.75 |
| InternVL2.5-4B (Chen et al., 2024b) | 61.43 | 61.54 | 36.40 | 58.41 | 21.99 | 32.14 | 32.43 | 73.87 | 71.56 | 69.47 | 57.56 | 59.15 | 41.21 | 54.52 |
| InternLM-XComposer2.5-7B (Zhang et al., 2024a) | 56.68 | 56.64 | 37.93 | 56.82 | 21.58 | 71.43 | 28.38 | 71.17 | 75.23 | 61.06 | 58.50 | 60.98 | 49.64 | 54.38 |
| InternLM-XComposer2-VL-7B (Dong et al., 2024a) | 59.18 | 52.45 | 40.23 | 56.91 | 25.31 | 66.07 | 31.08 | 67.57 | 73.39 | 61.06 | 52.34 | 53.66 | 57.15 | 53.80 |
| Hunyuan-Vision | 61.95 | 61.31 | 37.16 | 58.58 | 26.97 | 76.79 | 36.49 | 61.26 | 72.48 | 56.19 | 52.21 | 59.15 | 45.03 | 53.67 |
| Gemini-1.5-Pro (Reid et al., 2024) | 55.30 | 53.50 | 39.46 | 57.11 | 24.48 | 67.86 | 55.41 | 59.91 | 74.31 | 50.44 | 56.29 | 65.24 | 49.60 | 53.27 |
| Qwen2-VL-2B-Instruct (team, 2024) | 58.86 | 59.21 | 36.40 | 54.77 | 20.33 | 44.64 | 20.27 | 71.62 | 70.64 | 69.91 | 53.15 | 60.98 | 44.81 | 52.81 |
| InternVL2.5-2B-MPO (Chen et al., 2024b) | 60.95 | 62.47 | 37.16 | 53.40 | 36.93 | 30.36 | 32.43 | 68.02 | 68.81 | 63.72 | 53.28 | 58.54 | 36.31 | 52.65 |
| Mini-Gemini-34B (Li et al., 2023b) | 58.35 | 55.01 | 37.93 | 53.70 | 25.31 | 73.21 | 39.19 | 54.50 | 73.39 | 58.41 | 55.82 | 61.59 | 41.79 | 51.96 |
| Ovis1.6-Llama3.2-3B (Lu et al., 2024) | 59.79 | 51.98 | 41.00 | 53.14 | 24.07 | 39.29 | 45.95 | 74.32 | 77.06 | 68.58 | 52.21 | 62.80 | 32.79 | 51.64 |
| Molmo-7B-D (Deitke et al., 2024) | 68.02 | 55.71 | 37.16 | 52.40 | 24.90 | 48.21 | 40.54 | 56.76 | 67.89 | 46.02 | 53.41 | 60.98 | 42.70 | 51.61 |
| InternVL2-4B (Chen et al., 2024b) | 53.82 | 51.05 | 31.42 | 52.17 | 18.26 | 73.21 | 25.68 | 77.03 | 71.56 | 72.57 | 54.08 | 55.49 | 41.18 | 51.00 |
| MiniCPM-Llama3-V2.5 (Yao et al., 2024) | 51.93 | 50.12 | 36.40 | 49.88 | 19.92 | 76.79 | 20.27 | 69.37 | 77.06 | 68.14 | 56.49 | 62.20 | 41.79 | 50.95 |
| Mini-Gemini-34B-HD (Li et al., 2023b) | 54.95 | 47.09 | 37.55 | 48.87 | 27.80 | 73.21 | 40.54 | 59.91 | 72.48 | 58.85 | 58.37 | 66.46 | 35.91 | 50.35 |
| InternVL2.5-2B (Chen et al., 2024b) | 58.61 | 58.74 | 36.78 | 54.26 | 22.82 | 28.57 | 24.32 | 66.67 | 66.06 | 62.83 | 48.73 | 55.49 | 39.80 | 50.33 |
| Bunny-Llama-3-8B-V (He et al., 2024) | 58.16 | 51.05 | 34.87 | 54.07 | 21.58 | 50.00 | 12.16 | 45.95 | 66.06 | 53.10 | 48.73 | 57.32 | 59.44 | 49.93 |
| Mini-Monkey (Huang et al., 2024) | 52.25 | 56.64 | 26.82 | 52.53 | 26.56 | 73.21 | 18.92 | 68.92 | 65.14 | 59.29 | 56.60 | 50.00 | 42.37 | 49.46 |
| Phi3.5-Vision-Instruct (Abdin et al., 2024) | 55.01 | 45.69 | 30.27 | 52.61 | 21.16 | 66.07 | 31.08 | 45.05 | 63.30 | 53.10 | 53.95 | 53.66 | 54.65 | 49.12 |
| ColgVLM2-Llama3-Chat-19B (Hong et al., 2024) | 57.67 | 51.05 | 34.48 | 51.69 | 38.17 | 57.14 | 48.65 | 55.90 | 65.14 | 47.35 | 46.90 | 59.15 | 50.69 | 48.83 |
| Phi3-Vision-128K-Instruct (Abdin et al., 2024) | 55.30 | 39.86 | 30.27 | 51.61 | 25.31 | 69.64 | 40.54 | 45.05 | 65.14 | 47.79 | 45.25 | 60.37 | 56.75 | 47.57 |
| Yi-VL-34B (AI et al., 2024) | 53.02 | 38.23 | 30.27 | 50.33 | 26.14 | 64.29 | 17.57 | 50.45 | 56.88 | 55.31 | 51.00 | 52.44 | 53.88 | 47.38 |
| Step-1V-32K | 46.11 | 39.86 | 26.44 | 46.25 | 25.31 | 67.86 | 43.24 | 66.67 | 66.97 | 62.83 | 50.60 | 59.76 | 45.46 | 47.12 |
| Yi-VL-6B (AI et al., 2024) | 51.99 | 43.82 | 30.27 | 49.34 | 25.73 | 60.71 | 20.27 | 45.05 | 51.38 | 52.21 | 50.33 | 51.83 | 48.76 | 46.34 |
| ConvLLaVA-1024-7B (Ge et al., 2024) | 51.73 | 44.29 | 32.57 | 44.96 | 28.22 | 69.64 | 21.62 | 55.41 | 65.14 | 53.10 | 49.53 | 54.88 | 40.89 | 46.21 |
| ConvLLaVA-1536-7B (Ge et al., 2024) | 50.03 | 45.69 | 28.35 | 41.25 | 27.39 | 69.64 | 29.73 | 51.35 | 64.22 | 52.21 | 48.19 | 64.02 | 34.20 | 45.52 |
| Bunny-3B (He et al., 2024) | 49.97 | 49.65 | 26.82 | 48.79 | 25.73 | 50.00 | 12.16 | 46.40 | 61.47 | 47.79 | 42.17 | 51.22 | 55.08 | 45.40 |
| Bunny-4B-V1.0 (He et al., 2024) | 52.50 | 46.85 | 39.08 | 46.00 | 21.16 | 51.79 | 17.57 | 43.69 | 62.39 | 52.21 | 46.59 | 52.44 | 42.66 | 45.23 |
| LLaVA-HR-13B (Luo et al., 2024) | 50.32 | 41.26 | 35.25 | 39.81 | 32.37 | 66.07 | 27.03 | 44.50 | 60.55 | 45.58 | 48.46 | 57.32 | 48.80 | 45.12 |
| InternVL2-2B (Chen et al., 2024b) | 43.32 | 53.61 | 26.82 | 45.79 | 22.82 | 67.86 | 17.57 | 63.06 | 58.72 | 49.56 | 44.31 | 53.66 | 38.16 | 44.22 |
| Monkey-Chat (Li et al., 2024) | 49.20 | 47.55 | 24.14 | 47.32 | 16.60 | 69.64 | 13.51 | 51.35 | 58.72 | 44.25 | 44.18 | 51.22 | 48.91 | 44.10 |
| InternVL2.5-1B (Chen et al., 2024b) | 47.43 | 52.45 | 28.35 | 49.43 | 25.31 | 35.71 | 14.86 | 49.55 | 67.89 | 52.65 | 41.63 | 53.66 | 36.93 | 43.98 |
| InternVL2.5-1B-MPO (Chen et al., 2024b) | 45.05 | 51.05 | 28.35 | 49.29 | 29.46 | 25.00 | 35.14 | 48.65 | 64.22 | 48.67 | 42.95 | 56.71 | 34.71 | 43.12 |
| Molmo-7B-O (Deitke et al., 2024) | 54.72 | 47.79 | 36.78 | 40.54 | 23.24 | 39.29 | 28.38 | 47.75 | 64.22 | 46.46 | 42.30 | 60.37 | 31.59 | 43.05 |
| Mini-Gemini-13B (Li et al., 2023b) | 43.71 | 37.76 | 27.20 | 41.63 | 21.58 | 62.50 | 33.78 | 55.86 | 68.81 | 50.44 | 50.74 | 57.32 | 32.28 | 42.75 |
| SliME-7B (Zhang et al., 2024b) | 45.70 | 44.52 | 28.74 | 40.76 | 31.12 | 62.50 | 20.27 | 43.24 | 59.63 | 48.23 | 50.74 | 50.00 | 30.03 | 42.52 |
| INF-LLaVA* (Ma et al., 2024) | 43.19 | 44.76 | 32.95 | 41.92 | 24.48 | 57.14 | 20.27 | 50.00 | 66.06 | 55.31 | 45.65 | 54.27 | 31.41 | 42.41 |
| SliME-8B (Zhang et al., 2024b) | 46.50 | 41.72 | 32.18 | 40.27 | 30.29 | 60.71 | 25.68 | 44.59 | 61.47 | 47.79 | 50.87 | 47.56 | 29.96 | 42.39 |
| INF-LLaVA (Ma et al., 2024) | 45.66 | 40.09 | 27.59 | 47.37 | 33.20 | 57.14 | 32.43 | 46.40 | 60.55 | 42.92 | 41.63 | 54.88 | 35.58 | 42.18 |
| Mini-Gemini-13B-HD (Li et al., 2023b) | 42.29 | 34.73 | 32.18 | 40.20 | 18.67 | 67.86 | 24.32 | 51.35 | 63.30 | 45.58 | 46.99 | 56.71 | 34.28 | 41.99 |
| SliME-13B (Zhang et al., 2024b) | 47.46 | 40.33 | 28.74 | 42.55 | 22.41 | 66.07 | 17.57 | 45.50 | 56.88 | 47.79 | 45.52 | 56.10 | 33.55 | 41.73 |
| LLaVA-HR-7B (Luo et al., 2024) | 40.46 | 42.42 | 31.42 | 40.43 | 28.22 | 64.29 | 37.84 | 46.85 | 60.55 | 48.67 | 45.52 | 56.71 | 33.04 | 41.71 |
| ConvLLaVA-768-7B (Ge et al., 2024) | 46.50 | 39.16 | 28.35 | 40.88 | 16.60 | 66.07 | 22.97 | 53.15 | 69.72 | 54.42 | 45.65 | 55.49 | 37.11 | 41.51 |
| InternVL2-1B (Chen et al., 2024b) | 43.13 | 46.85 | 22.99 | 43.29 | 23.24 | 64.29 | 18.92 | 54.05 | 58.72 | 49.12 | 44.44 | 57.93 | 27.89 | 41.41 |
| DeepStack-L-HD-Vicuna-7B (Meng et al., 2024) | 43.29 | 34.97 | 28.74 | 35.74 | 18.67 | 60.71 | 17.57 | 46.85 | 60.55 | 45.13 | 42.97 | 59.15 | 35.88 | 39.21 |
| mPLUG-Owl2 (Ye et al., 2024) | 40.04 | 36.83 | 28.74 | 42.93 | 26.97 | 30.36 | 12.16 | 41.89 | 60.55 | 38.94 | 44.58 | 50.61 | 30.36 | 38.77 |
| DeepStack-L-Vicuna-7B (Meng et al., 2024) | 45.47 | 39.16 | 27.20 | 36.00 | 21.99 | 60.71 | 18.92 | 42.34 | 56.88 | 42.04 | 42.97 | 50.61 | 30.21 | 38.60 |
| Random Choice | 23.12 | 24.01 | 21.84 | 25.85 | 29.46 | 35.71 | 25.68 | 36.94 | 46.79 | 38.50 | 34.00 | 47.65 | 28.61 | 29.90 |