# OpenReview forum: "MMCOMPOSITION: Revisiting the Compositionality of Pre- trained Vision-Language Models"
_TMLR — Accepted by TMLR_

### Review · Reviewer_Temx · 2026-03-03

**Summary Of Contributions:**

This paper introduces MMCOMPOSITION, a novel benchmark to evaluate the performance of VLMs across complex visual compositional and reasoning tasks. The authors state that existing benchmarks are not able to capture the measurement of fine-grained vision-language compositional understanding. The benchmark contains almost 4.1k questions and the authors evaluate 77 VLMs across various architectures like Qwen, LLava and closed source models liek GPT-4o and Gemini 1.5. The authors also conduct ablations on varying the input image resolution, training data, and language decoder size.

Strengths:
1. The paper adds 13 sub-categories across perception, reasoning and probing like attribute perception, object perception, object counting, difference counting, etc. This allows for more precise evaluation of VLMs and clear attribution of which areas of a particular model need to be improved.
2. The benchmark is of high quality as it is human annotated and verified and the authors also report a human score of 90.31%
3. The authors conduct a massive evaluation of 77 VLMs, including both open-source models and closed source API based models.

Weaknesses:
1. The authors do not include any of the latest models SoTA models like GPT-5 - GPT5.2, Claude Opus/Sonnet 4.5 - 4.6, and Gemini 3 or 3.1. The authors run the evaluations on GPT-4o and Gemini 1.5 which are around 1.5 years old. It is important to evaluate atleast a couple of these models to understand the difficulty of the benchmark and how useful it might be for the future series of models.
2. Leakage risk: The benchmark uses as seed QA pairs several existing datasets in the VL domain like NLVRv2, Visual Genome, etc. These datasets and seeds are frequently found in pre-training splits of existing VLMs so its important to do a deduplication check and make sure that the evaluations are done while accounting for these possible leakages.

**Audience:**

Yes

**Audience Explanation:**

Researchers working in the field of VLMs, reasoning and fine-tuning of VLMs will find this benchmark useful for their use cases.

**Claims And Evidence:**

Yes

**Claims Explanation:**

The authors include detailed experiments on their benchmark, comparing the performance of around 77 VLMs and also include detailed information about how the benchmark was constructed and which seed datasets were used to create the benchmark. They also include detailed ablations across language decoder size, training data, and input image resolution.

**Requested Changes:**

Please refer to the weaknesses section above. In particular, I feel the below are important to have recommendation of accept:

1. Running baselines on latest SoTA models like some of GPT-5 - GPT5.2, Claude Opus/Sonnet 4.5 - 4.6, and Gemini 3 or 3.1. It is important to check the validity and difficulty of the benchmark against these models.

2. Deduplication check and analysis: Investigate which datasets from the seed datasets are a part of VLMs training data and dedup and run the evaluations again to get a fair metric for the different models.

---

### Review · Reviewer_tucA · 2026-03-03

**Summary Of Contributions:**

This paper delivers a unified, hard-sample–filtered VLM evaluation suite spanning diverse tasks, domains, and capabilities.

It introduces a rule-based conversion pipeline that standardizes heterogeneous sources into a QA format (retrieval-based distractors, multi-image concatenation, indefinite-choice). Besides generating eval data, this pipeline can also be repurposed to create training data.

Extensive experiments across many open and proprietary VLMs show the benchmark is challenging, reveal a substantial human–model gap, and report breakdowns by category, difficulty, and multi-hop.

**Audience:**

Yes

**Audience Explanation:**

1. A unified, diverse, and hard-filtered evaluation suite is highly valuable in practice for stress-testing VLMs during development, often more efficient than juggling many source benchmarks separately.
2. The paper’s rule-based question/option construction pipeline (e.g., distractor retrieval, multi-image formatting, indefinite-choice probing) could plausibly be repurposed for training data curation and augmentation, which is of interest beyond evaluation alone.

**Claims And Evidence:**

No

**Claims Explanation:**

* **Key concepts are not operationalized**: The paper frames the benchmark as measuring “compositionality” and emphasizes “multi-hop,” but it does not provide a precise, reproducible definition of these concepts at the sample level. As a result, it is unclear what type of “compositionality” capability is actually being measured beyond a mix of “hard” VLM questions from multiple hypertasks.
* **Multi-hop claims lack traceable identification and labeling evidence**: The paper reports that 2,371 samples are multi-hop and uses Table 8/11 to build a narrative around multi-hop difficulty, yet it does not explain how multi-hop samples were generated or identified—e.g., whether they come directly from multi-hop benchmarks such as CLEVR, from human annotation, or are induced by the rule-based QA/option construction pipeline. The paper also does not specify a decision rule for labeling an evaluation sample as multi-hop. Without this, the multi-hop subset analysis is difficult to interpret and hard to trust.
* **Human-annotation is repeatedly emphasized but not documented**: The submission uses “human-annotated” as an implicit quality guarantee, but does not provide an annotation protocol: what annotators actually did (write QA vs verify vs rewrite), how many annotators, coverage rate, guidelines (e.g., “aspect prompts”), disagreement resolution, or error/ambiguity rates. This weakens the credibility of claims about benchmark quality and correctness.
* **Construction pipeline details are insufficient to validate the benchmark or rule out bias**: The rule-based QA/option construction (retrieval-based distractors, multi-image concatenation, manually written misaligned captions) is a major part of the contribution, but key implementation and quality-control details are missing (retrieval filtering, negative option constraints, ambiguity control, semantic fidelity checks, deduplication). Since these design choices can materially change difficulty and failure modes, the evidence is not strong enough to support broad claims about “compositional” capabilities.
* **Evidence tables are largely leaderboard-style and do not substantiate the stronger interpretive claims**: Although the paper reports extensive results (e.g., Figure 2), they are presented primarily as large score matrices without clear structure or controlled analysis to support the takeaways.
* **Missing reasoning-oriented baselines**: Since the paper emphasizes “compositionality” and “multi-hop,” it would be important to include at least one strong reasoning-oriented model as a baseline. While continuously updating to the latest models (e.g., Gemini 3, Qwen-3, GPT-5) may be impractical, the absence of a reasoning-focused baseline limits the conclusions—especially because such models might better infer and exploit the latent rules underlying the benchmark’s QA/option construction pipeline, potentially reducing the apparent difficulty in a way not captured by the current model set.

**Requested Changes:**

* Operationalize “compositionality” and “multi-hop,” and make multi-hop labeling reproducible: Provide a clear, sample-level definition of compositionality and multi-hop, along with a transparent labeling rule or pipeline. Explicitly describe how the 2,371 multi-hop samples are identified and provide a breakdown of their sources.
* Document human annotation with quantitative protocol details: Clarify what “human-annotated” means in practice. Specify how many samples are human-verified or human re-labeled, and briefly describe the annotation procedure.
* Clarify the QA/option construction pipeline and audit label correctness: Provide sufficient detail on the rule-based transformation process, including concrete templates/prompts used to pack options into MCQ or indefinite-choice formats. Explain how newly generated negative options are validated (e.g., human verification or automatic checks) to ensure QA/option correctness.
* Reorganize experimental results to support interpretable trends: For large tables (e.g., Table 2), group or curate models in a principled manner (e.g., by model family, scale, or training regime) and include concise summaries to highlight key trends. A smaller, more focused set of representative models may be more informative than a large raw leaderboard matrix.
* Add at least one reasoning-oriented baseline: Since the benchmark emphasizes compositionality and multi-hop reasoning, include results from at least one strong reasoning-focused model (or a clearly configured “thinking mode” baseline) to better contextualize the difficulty and strengthen comparative conclusions.
* Improve presentation clarity: Fix figure and table captions and formatting issues (e.g., missing captions such as in Figure 4 of the appendix, ambiguous model column in Table 11), and ensure appendix references and links are complete and consistent.

---

### Review · Reviewer_ea7m · 2026-03-12

**Summary Of Contributions:**

This paper introduces MMComposition, a human-annotated benchmark of 4,122 questions designed to evaluate how well vision-language models understand compositional visual information. Existing compositionality benchmarks like ARO and Winoground are limited to image-text retrieval tasks and only assess basic attribute, relation, and object recognition, overlooking deeper reasoning about object interactions, counting, and complex multi-image compositions. Broader VLM benchmarks like MMBench, MMMU, and MMVet evaluate general capabilities but are not specifically designed for fine-grained compositional evaluation. MMComposition addresses these gaps by covering 13 categories across perception, reasoning, and probing, and by introducing both single-image and multi-image scenarios with single-choice and indefinite-choice question formats — a much richer evaluation setup than prior work. The authors benchmark 77 models and find that the best (Qwen2.5-VL-72B, 68.16%) still lags far behind human experts (90.31%), with models particularly struggling on counting, difference spotting, and multi-answer probing tasks. A systematic analysis of architectural factors reveals that the visual encoder is more important than language model size for compositional understanding once the decoder is sufficiently large (34B+), which explains why some open-source models outperform GPT-4o — whose image downsampling degrades fine-grained visual information.

**Audience:**

Yes

**Audience Explanation:**

The paper addresses a relevant gap in VLM evaluation by systematically benchmarking compositional understanding, a core capability that existing benchmarks don't adequately cover. The diagnostic analysis of how visual encoders, decoder size, and training data affect compositionality provides actionable insights that can be useful to researchers building or improving VLMs.

**Broader Impact Concerns:**

The seed images (as discussed in 'Data Collection' paragraph in page 5) should be available under proper license for open-sourcing, and be used for research. I request the authors to cross-validate.

**Claims And Evidence:**

Yes

**Claims Explanation:**

I would explain my rating by listing the strengths and weaknesses of the manuscript, and suggest the authors to respond to the mentioned major weaknesses.

**Strengths**

(1) The paper addresses an important topic of compositionality in VLMs, and explains the limitations of existing benchmark in this field. The paper is overall well-written, easy to read, and the data curation pipeline and experiments are well-designed.

(2) The experiments systematically analyze the importance of *(i)* high-resolutional visual encoder, *(ii)* mixer-of-encoders, *(iii)* language decoder size and *(iv)* the volume of training data. Moreover, the appendix Table 13 shows an extensive performance comparison of 77 VLMs, which sets a strong benchmarking of the proposed dataset.

(3) During the data curation, the negative options in MCQs are chosen with a controlled system by retrieving the most similar negative captions. The dataset contains several tasks containing two or more than two images, which evaluates the model's fine-grained multi-image understanding and comparison capability. The VL composition probing task also contains multiple correct answers in MCQs, which is a significantly more difficult task than single-correct answer QAs.

(4) The supplementary material contains a good amount of qualitative examples.


**Major Weaknesses**

(1) Issues with data curation:

(a) The naming of 'infinite-choice' for probing type of questions is somewhat misleading as there are always 4 given options. A better naming can be 'single-correct MCQ' vs 'multi-correct MCQ'.

(b) The authors did not explain the reasoning behind choosing the 13 category of questions. An extension of Table 1, detailing which existing datasets contain which subset of these 13 categories would provide a clear comparison and novelty of the proposed benchmark.

(c) During the negative option curation, the authors mention: 'We then utilize these embeddings to retrieve the most similar captions from the Visual Genome (Krishna et al., 2016) dataset.' -- it is not clear to me why Visual Genome dataset is used to select the negative captions? Moreover, only selecting the semantic embedding for choosing visual caption may not be the best method, because the model under evaluation can reject the given choices just by filtering negative object names, rather than truly understanding their spatial relationships.

(d) For the probing task, why are the misaligned captions manually written rather than following the same technique as (c)?

(e) The authors mention that 'over half of the data containing more than four options', but it is not clear which question categories contain 5 options, specially because all the examples shown in Figure 1 have two or four options.

(2) Issues with benchmarking and training:

(a) First of all, I am confused which set of experiments require model training? I guess the main results in Table 2 are evaluation only, but did the authors retrain the models in Table 3 and Table 4? It is very important to clarify upfront which models are retrained, and what data is used for training, the all related hyper-parameters, i.e. epochs, learning rate, training schedule, and required compute infra, cost.

(b) In the Table 4 (mixture-of-expert comparisons), 'LLaVA-1.5+A+B'  and 'LLaVA-1.5+A+B+C' models perform worse than 'LLaVA-1.5' and 'LLaVA-1.5+A', but 'LLaVA-1.5+A+B+C+D' performs better than 'LLaVA-1.5+A+B' and 'LLaVA-1.5+A+B+C'.  Do the authors have clear explanation for such behavior? Is there a specific reason why the additional encoders are added in this specific order?

(c) I also request the authors to provide results with the latest VLMs, such as Gemini-3-Pro, and GPT-5.4 thinking models in the updated version of the manuscript.

(3) The paper would also benefit from explaining a few failure modes from the strongest model, and understand where these models still significantly lack compared to human performance.


**Minor Weaknesses**

(1) The difficulty classification approach is not convincing enough, as the used models are small-scaled, and they did not perform well in appendix Table 13. I do not clearly understand the purpose of the difficulty classification, specially when such a rating is subjective. If it needs to be done, a cleaner human-annotation is required.

(2) Figure 2, being the primary figure showing the statistics of the proposed benchmark, is hard to read (specially for color-blind people). The fonts needs to be increased, the the contrast can be designed better.

(3) In the 'Human Annotation' paragraph in Page 6, the authors mention that 'Annotators first assess image quality to ensure it meets the required standards.' Is there a designed guidelines for defining the 'required standard'? Without a clear guideline, the meaning of 'required standard' is a subjective entity.

**Requested Changes:**

Overall, the proposed dataset may have the potential to be valuable to the community. I request the authors to address my comments in the **Major Weaknesses** section, and update the manuscript accordingly.

---

### Decision · Action_Editor_Npch · 2026-04-24

**Recommendation:** Accept as is

**Additional Comments:**

Reviewers are overall happy with the improvements to the manuscript, e.g. operationalizing compositionality versus multi-hop and changing the language of "infinite-choice". A few small remaining requests from reviewers include:
* Continuing to improve the definition of compositionality, potentially with a sample
* Add descriptions of single-hop versus multiple-hop with 1-2 concrete examples
* Softening the wording of some claims
* Running sanity check (perhaps just using authors) to bolster findings

**Audience:**

Yes

**Audience Explanation:**

The TMLR audience will be interested in this paper's findings. Two reviewers do note the paper is good but the novelty may be slightly more limited.

Based on the overall reviews and state of the field, I believe MMComposition and its findings will be most interesting to researchers analyzing visiolinguistic perception and reasoning, VLM training (due to the diagnostic analyses in Section 5) and benchmark creation because of the pipeline setup.

**Claims And Evidence:**

Yes

**Claims Explanation:**

This paper's claims are supported by accurate, convincing and clear evidence.

General strengths of the paper (with findings described) are:
* *structuring compositionality into categories:* This work deepens existing knowledge on compositionality by structuring compositionality into 13 subcategories (Reviewer Temx). It goes deeper than prior work that often focuses primarily on object relations, attributes, etc. With these categories, they are to determine that models generally perform best on object, attribute and relation perception and reasoning.
* *extensive evaluation:* The paper tested a large number of models to ensure the results generalize (Reviewer Temx, Reviewer ea7m). Given the extension number of models tested, the authors also do a deeper probe and explore dimensions like the impact of the language decoder size, resolution of the vision encoder and amount of training data. They find, for instance, that like prior work scaling the training data can lead to up to 10x improvement in compositional understanding.
* *solid pipeline for benchmark creation:* The pipeline to construct MMComposition could be repurposed (Reviewer tucA), especially in how it tests multiple dimensions in a single unified evaluation instead of needing multiple benchmarks. This shows the pipeline is clean and generalizable to other sub-areas within the field.


Overall, I find that MMComposition is a strong benchmark that can be used to probe VLMs along with a solid pipeline for dataset construction.

---

> ### Author Response · Authors · 2026-04-25
> **Thank You for the Review and Decision！**
>
> Dear Action Editor and Reviewers,
>
> Thank you very much for your time, careful evaluation, and thoughtful feedback throughout the review process. We sincerely appreciate the positive assessment and are delighted that the paper is recommended for acceptance!
>
> We are also grateful for the remaining suggestions, including further clarifying the definition of compositionality, adding concrete examples for single-hop versus multi-hop reasoning, softening several claims, and including a sanity check to further support the findings. We will carefully incorporate these improvements in the final version to further strengthen the paper.
>
> Thank you again for your support and constructive comments.
>
> Best,
>
> Authors